# Engineered Glucose Oxidase Capable of Quasi-Direct Electron Transfer after a Quick-and-Easy Modification with a Mediator

**DOI:** 10.3390/ijms21031137

**Published:** 2020-02-08

**Authors:** Nanami Suzuki, Jinhee Lee, Noya Loew, Yuka Takahashi-Inose, Junko Okuda-Shimazaki, Katsuhiro Kojima, Kazushige Mori, Wakako Tsugawa, Koji Sode

**Affiliations:** 1Department of Biotechnology and Life Science, Graduate School of Engineering, Tokyo University of Agriculture and Technology, 2-24-16 Naka-cho, Koganei, Tokyo 184-8588, Japan; nsuzuki@protonmail.com (N.S.); tsugawa@cc.tuat.ac.jp (W.T.); 2Joint Department of Biomedical Engineering, The University of North Carolina at Chapel Hill and North Carolina State University, Chapel Hill, NC 27599, USA; jh.lee@unc.edu (J.L.); noya-loew@rs.tus.ac.jp (N.L.); jokudas@email.unc.edu (J.O.-S.); 3Ultizyme International Ltd., 3-9-5. Taihei, Sumida, Tokyo 130-0012, Japan; yukappe1122@gmail.com (Y.T.-I.); katsuhiro.kojima@gmail.com (K.K.); mori_ka1213@yahoo.co.jp (K.M.)

**Keywords:** glucose oxidase, direct electron transfer, amine-reactive phenazine ethosulfate, glucose sensor, glycemic level monitoring

## Abstract

Glucose oxidase (GOx) has been widely utilized for monitoring glycemic levels due to its availability, high activity, and specificity toward glucose. Among the three generations of electrochemical glucose sensor principles, direct electron transfer (DET)-based third-generation sensors are considered the ideal principle since the measurements can be carried out in the absence of a free redox mediator in the solution without the impact of oxygen and at a low enough potential for amperometric measurement to avoid the effect of electrochemically active interferences. However, natural GOx is not capable of DET. Therefore, a simple and rapid strategy to create DET-capable GOx is desired. In this study, we designed engineered GOx, which was made readily available for single-step modification with a redox mediator (phenazine ethosulfate, PES) on its surface via a lysine residue rationally introduced into the enzyme. Thus, PES-modified engineered GOx showed a quasi-DET response upon the addition of glucose. This strategy and the obtained results will contribute to the further development of quasi-DET GOx-based glucose monitoring dedicated to precise and accurate glycemic control for diabetic patient care.

## 1. Introduction

Glucose oxidase (β-d-glucose: oxygen 1-oxidoreductases, E.C. 1.1.3.4, GOx) is a flavoprotein that catalyzes the oxidation of β-d-glucose at its hydroxyl group linked to the carbon one using molecular oxygen as the electron acceptor to produce d-glucono-1,5-lactone and hydrogen peroxide (Scheme 1). Since the first report by Clark and Lyons on an enzyme electrode employing GOx and an oxygen electrode for glucose monitoring [1], extensive studies have been carried out to develop improved enzyme-based systems for monitoring glycemic levels. The reason for the popularity of GOx lies in its commercial availability, high activity, and substrate specificity against glucose.

Electrochemical glucose biosensors can be divided into three principles (Scheme 2). The “first-generation” glucose sensing principle utilizes molecular oxygen as the primary electron acceptor, and the glucose concentration is determined by the consumption of oxygen or generation of hydrogen peroxide. In the “second-generation” principle, artificial electron acceptors (also referred to as electron mediators) replace oxygen in the reaction [2], which enables avoiding the production of harmful hydrogen peroxide. Furthermore, due to the lower redox potential compared to that of hydrogen peroxide, the use of certain mediators enables us to decrease the applied potential for biosensors based on glucose oxidase. However, when oxidase is employed, this reaction is inherently affected by the oxygen concentration in a sample. Accordingly, glucose dehydrogenase (GDH) has become the major enzyme for glucose sensing in the second-generation principle. However, because of GDH’s broad substrate specificity, which can lead to potentially fatal errors in glucose sensing [3], GOx has still been the focus due to its high specificity. Due to the advantages of GOx over GDH, studies are ongoing to engineer GOx into dehydrogenase [4,5,6,7]. In the “third-generation” sensor principle, glucose dehydrogenases capable of direct electron transfer (DET) are employed. Thereby, electrons derived from glucose oxidation are transferred directly to the electrode. Glucose sensor employing DET-type enzymes has several advantages comparing mediator type glucose sensors, e.g., a smaller number of essential elements for sensing, a smaller number of essential reactions for detection, operation at a lower applying potential than those utilizing hydrogen peroxide, and a major kind of mediator used in second-generation principle such as ferrocene, ferricyanide, and ruthenium complexes [8,9]. This low-applying potential contributes to eliminate interference by electrochemically active ingredients in the samples. Therefore, the DET-based enzyme sensor principle has been considered ideal for redox enzyme-based electrochemical sensors.

Considering the superior enzymatic property of the “gold standard,” several efforts have been made to realize DET GOx-based glucose sensing. However, the 3D structure of GOx revealed that the redox cofactor, flavin adenine dinucleotide (FAD) is buried deeply within the protein. In addition, previous reports [10,11] attributed the voltammograms associated with GOx to free FAD, which suggests that only the free FAD of GOx showed voltammetric signals, whereas bound FAD did not, and bound FAD undergoes electron transfer at a more positive potential not observed in voltammetry [12]. Therefore, the only approach that realizes DET for GOx-based glucose sensing is the modification of the redox species onto the enzyme surface. Since the 1980s, GOx modifications by covalently attaching redox mediators have been reported to achieve quasi-DET using derivatives of ferrocene [13,14,15,16,17,18,19,20,21], phenothiazine [22], ruthenium [23], tetrathiafulvalene [24], and phenoxazine [25] to modify GOx. However, the modification procedure requires complex and laborious steps to introduce functional groups for mediator modification. In addition, no report has demonstrated the rational design of residues where redox mediators should be modified on GOx.

To achieve a versatile and simple method for enzyme modification, we previously reported a rapid and single-step conjugation using 1-[3-(succinimidyloxycarbonyl)propoxy]-5-ethylphenazinium trifluoromethanesulfonate (or amine-reactive phenazine ethosulfate; arPES) [26,27] onto the surface of the redox enzyme. The attachment of these redox mediators to redox enzymes enables intramolecular electron transfer from the enzyme redox center to the mediator and intermolecular electron transfer from the mediator to an electrode. This electron transfer via the mediator bound to the enzyme surface can be described as quasi-direct electron transfer (quasi-DET) and “2.5th generation” principle. arPES harbors two parts, i.e., a succinimide group and mediator, namely, methoxy PES, which is a stable, relatively low redox potential mediator. Upon mixing arPES and an enzyme, the primary amine groups on the enzyme attack the carboxylate ester carbon of the succinimide group of arPES and form amide bonds at room temperature.

Therefore, in this study, we aimed to design an engineered GOx suitable for quasi-DET-type sensor applications through the quick and easy modification of mediators using arPES. As the enzyme, GOx from *Aspergillus niger* was chosen, and amino acid substitution was designed based on the 3D structure to introduce a lysine residue in GOx, which was modified by arPES. After investigating the characteristics of the engineered GOx, PES-modified GOx was prepared, and a quasi-DET-type glucose sensor was constructed.

## 2. Results and Discussion

### 2.1. Engineering of GOx for PES Modification

#### 2.1.1. Investigation of the Availability of Wild-Type *Aspergillus niger*-Derived GOx for Redox Mediator Modification

First, we investigated whether wild-type GOx from *A. niger* (*An*GOx-WT) is readily available for modification by PES to show a quasi-DET response. *An*GOx-WT was modified with PES, and the dye-mediated dehydrogenase activity was determined with an electron mediator, phenazine methosulfate (PMS), and a terminal electron acceptor, 3-(4,5-dimethyl-2-thiazolyl)-2,5-diphenyl-2*H*-tetrazolium bromide (MTT). When PMS and MTT were used together (PMS/MTT system), the PMS functioned as the primary electron acceptor of the enzyme and mediated electrons to the bulky MTT. We previously reported that MTT alone did not function as the electron acceptor for a variety of redox enzymes. However, the presence of PMS facilitated the reduction of MTT by the oxidation substrate [26]. Considering that both PMS and MTT are negatively charged in their oxidative status, we assumed that the low reactivity of MTT as the primary electron acceptor might be attributed to its bulkiness when compared with PMS. Therefore, we have used the availability of MTT as the sole electron acceptor in the solution (MTT system) as the indication that PES was successively modified on the surface of redox enzymes, where internal electron transfer from a deeply buried cofactor to PES occurred. Consequently, bulky MTT can be reduced [26]. Since GOx was revealed that it did not utilize MTT as the primary electron acceptor, we followed our previous empirical experimental condition to analyze the modification of PES on the surface of GOx and evaluate the availability of the arPES modification procedure to prepare quasi-DET type GOx. We assumed that successive PES modification might enable PES-modified GOx to use MTT as the primary electron acceptor to show dye-mediated glucose dehydrogenase activity.

The results are shown in Table 1. The PES-modified *An*GOx-WT revealed almost identical dye-mediated dehydrogenase activity as that of unmodified and PES-modified *An*GOx-WT when the PMS/MTT system was employed, which indicates that the PES-modified GOx maintained its original activity. However, in the MTT system, only low dehydrogenase activity was observed after the modification of PES, which was indistinguishable from that of the intact *An*GOx-WT.

To investigate the presence of the PES molecule on the surface of *An*GOx-WT after PES modification, we further characterized the enzyme on an electrode. The cyclic voltammograms of PES-modified *An*GOx-WT clearly showed oxidation and reduction peaks at approximately -100 mV, which revealed the presence of PES on the surface of the enzyme (Appendix A), which were not observed from the electrode with PES-unmodified intact *An*GOx-WT (Appendix A). These results indicate that PES was successfully modified on the surface of GOx and was electrochemically active. Considering that no catalytic current was observed in the presence of glucose (Appendix A), together with the results of the previously mentioned dye-mediated dehydrogenase activity investigations using the MTT system, the position of PES modification on *An*GOx-WT was not suitable to accept electrons from its redox cofactor, FAD, in the catalytic center.

#### 2.1.2. Identification of the Appropriate Site for Substitution with a Lysine Residue

To locate PES at a suitable position for the electron relay, we designed an additional lysine residue, where arPES should make a covalent bond. To determine the position, we compared the structure of *An*GOx (PDB ID: 1CF3) with a model structure of *Botryotinia fuckeliana*-derived GDH (*Bf*GDH), which has a structure similar to that of *An*GOx but shows a quasi-DET response after PES modification. A closer look at the position of lysine residues in the crystal structures of *An*GOx (1CF3), a structural model of *Bf*GDH, and the crystal structure of *A. flavus*-derived GDH (*Af*GDH, PDB ID: 4YNU), on which the model was based, shows that the residues are distributed very differently, despite the similar overall structure (Figure 1). However, there is one notable residue that is located at what appears to be the entrance of a pathway to the catalytic center. This residue is a lysine in both GDHs (K483 in *Bf*GDH, K477 in *Af*GDH) and an isoleucine in *An*GOx (I489). Furthermore, this residue appears to be located close enough to the FAD cofactor to receive electrons. Therefore, this residue appears to be an ideal candidate for attaching PES to function as an electron relay between the FAD and the terminal electron acceptor.

### 2.2. Construction and Characterization of the GOx Mutant for Quasi-DET after PES Modification

The isoleucine residue I489 in *An*GOx was substituted with a lysine residue in the *An*GOx-I489K mutant. Kinetic parameters of the mutant are shown in the Appendix A (Appendix A). Notably, the modification of *An*GOx-I489K with PES significantly increased the activity value obtained with the MTT system (Table 1), which indicated the successful formation of an electron relay from FAD in the catalytic center to the enzyme surface. Additionally, a similar *K*_m_ value was observed with *An*GOx-I489K dehydrogenase activity before and after PES modification (Appendix A). Therefore, it appears that the attachment of PES to the substrate entrance did not alter the accessibility of glucose to the active center.

Cyclic voltammograms of PES-modified *An*GOx-I489K clearly showed oxidation and reduction peaks at approximately −100 mV, which indicates the presence of PES on the enzyme surface (Figure 2c). In the presence of glucose, a catalytic current, i.e., an increase in the current at potentials higher than that of the oxidation peak of PES, was observed when we used PES-modified *An*GOx-I489K (Figure 2d), and no such catalytic current was observed with unmodified *An*GOx-I489K (Figure 2b). The observed reduction peak decrease at −0.4 V upon addition of glucose, which would be attributed to the decrease of local oxygen concentration caused by oxygen consumption by immobilized GOx. This result confirms that *An*GOx-I489K acquired quasi-DET ability after modification with PES. PES bound to the lysine residue at position 489 provided the missing link in the electron relay from FAD to the electrode.

### 2.3. Characterization of Electrodes with PES-Modified AnGOx-I489K

Lastly, we investigated the quasi-DET response against various glucose concentrations. Chronoamperometric measurements using the electrode with immobilized PES-modified *An*GOx-I489K showed clear and stable responses for the addition of glucose at an operating potential of 0 mV vs. Ag/AgCl (Figure 3A). A plot of the response current against the glucose concentration showed a linear increase until approximately 3 mM and saturation at glucose concentrations higher than approximately 17 mM (Figure 3B). The response on the electrode gave the apparent *K*_m_ value of 2.3 mM and *I*_max_ value of 1.4 µA even though the mutation nor PES modification did not change the *K*_m_ value of GOx (Appendix A). This smaller apparent *K*_m_ value observed in the electrochemical measurement was assumed to be the difference of a rate limiting step. In the homogeneous solution, the reductive half reaction of the cofactor was the rate limiting step. However, in the electrochemical measurement, the electron transfer between quasi-DET GOx and the electrode might be limited. Essentially, the rate limiting step in the electrochemical reaction with PES modified GOx was assumed at the oxidative half reaction where reduced PES was oxidized by the electrode. Therefore, the apparent *K*_m_ value observed on the electrode did not match the actual *K*_m_ value determined with the redox dye.

The cyclic voltammograms (Figure 2) and chronoamperometric measurements (Figure 3A,B) confirm that PES-modified *An*GOx-I489K is capable of quasi-direct electron transfer due to PES modification of the introduced lysine residue. Based on these results, it seems that, after the FAD cofactor is reduced by glucose, it is re-oxidized by the PES bound to the lysine residue at position 489. The reduced PES is then oxidized directly at the electrode or possibly via other PES molecules bound at the surface of GOx.

## 3. Materials and Methods

### 3.1. Materials

arPES was kindly donated by Dojindo Laboratories Co. Ltd. (Kumamoto, Japan). Phenazine methosulfate (PMS), d(+)-glucose, potassium dihydrogen phosphate, dipotassium hydrogen phosphate and Triton X-100 were purchased from Kanto Chemical Co. Inc. (Tokyo, Japan). 3-(4,5-Dimethyl-2-thiazolyl)-2,5-diphenyl-2*H*-tetrazolium bromide (MTT) was purchased from Dojindo Laboratories Co. Ltd. d-(+)-trehalose dihydrate and glutaraldehyde were purchased from FUJIFILM Wako Pure Chemical Corporation (Osaka, Japan). *N*-[Tris(hydroxymethyl)methyl]glycine (tricine) was purchased from Sigma-Aldrich Co. LLC (St. Louis, MO, USA). Ketjen black (ECP600JD), was obtained from Mitsubishi Chemical Corporation (Tokyo, Japan).

### 3.2. Identification of the Appropriate Mutation Site by Comparing the GOx and GDH Structures

A 3D structural model of *Botryotinia fuckeliana*-derived GDH (*Bf*GDH) was generated based on the crystal structure of FAD-dependent glucose dehydrogenase derived from *A. flavus* (*Af*GDH, PDB ID: 4YNU) [28]. The amino acid sequences of GOx derived from *A. niger* (*An*GOx, PDB ID: 1CF3) and of the GDHs were aligned using the multiple sequence alignment software ClustalW (http://www.clustal.org) [29]. By comparing the structures of GOx and the GDHs, lysine residues in the GDHs were identified, which are close to what appears to be the entrance of pathways leading to the catalytic center and which have corresponding non-lysine residues in the GOx.

### 3.3. Expression Vector Preparation and Recombinant Expression of GOxs

The structural gene of wild-type GOx was prepared as previously described [7]. The gene was inserted into the multiple cloning site of the expression vector pET30c(+) (Merck KGaA, Darmstadt, Germany). Expression vectors of GOx mutants were prepared by site-directed mutagenesis using the QuikChange Mutagenesis Kit (Agilent Technology Inc., Santa Clara, CA, USA). The correctness of the mutations was confirmed using the ABI Prism 3100 Genetic Analyzer (Applied Biosystems, Foster City, CA, USA). Expression and refolding of GOxs (both wild-type and mutant) were carried out as previously described [30].

### 3.4. PES Modification of Enzymes

GOxs (wild-type and mutant) were modified using arPES, as previously described [26] with some changes. GOx and arPES were mixed in 20 mM tricine buffer (pH 8.3) to 13 μM (based on catalytic centers) and 2.0 mM, respectively, and incubated at 25 °C for 2 h while shaking at 1200 rpm. Afterward, the sample was ultra-filtered to remove excess arPES and exchange the buffer to 20 mM potassium phosphate buffer (P.P.B.) using Amicon Ultra-0.5 30K centrifugal filters (molecular cut-off, 30 kDa) (Sigma-Aldrich Co. LLC, St. Louis, MO, USA).

### 3.5. Enzyme Activity

Two types of dehydrogenase activity assays were performed, which were named the PMS/MTT system and MTT system. For the PMS/MTT system, the increase in absorbance at 565 nm (from the formation of formazan dye due to the reduction of MTT) of a mixture of GOx, 100 mM glucose, 0.6 mM PMS, and 1 mM MTT in 20 mM P.P.B. was monitored. For the MTT system, the increase in absorbance at 565 nm of a mixture of GOx, 100 mM glucose, and 1 mM MTT in 20 mM P.P.B. was monitored. The reduction of 1 μmol MTT in 1 min, corresponding to the oxidation of 1 μmol/min glucose, was defined as 1 U dehydrogenase activity.

### 3.6. Preparation of Enzyme Electrodes

First, 0.54 μL of enzyme ink containing 1.5 mg/mL GOx (unmodified or PES-modified), 0.5% trehalose, 0.6% Ketjen black, and 1.2% Triton X-100 in 20 mM P.P.B. was dropped on a glassy carbon electrode (3.0 mm diameter, BAS Inc., Tokyo, Japan), and dried at 25 °C. Next, the enzyme was cross-linked in glutaraldehyde vapor for 1 h at room temperature. The electrodes were stored at 25 °C until use.

### 3.7. Electrochemical Evaluation

Before use, the electrodes were equilibrated in the electrolyte (100 mM P.P.B., pH 7.0) for at least 20 min. A platinum wire (TANAKA Kikinzoku K.K., Tokyo, Japan) was used as a counter electrode and Ag/AgCl/3 M NaCl as a reference electrode (BAS Inc., Tokyo, Japan). An SP-150 potentiostat (Bio-Logic Science Instruments, Seyssinet-Pariset, France) was used for electrochemical measurements. Cyclic voltammetry was performed for a potential range of −0.7 V to 0.3 V at a scan rate of 50 mV/s. For the chronoamperometric measurements, a potential of 0 V was applied. Glucose was added to the final concentrations of 0, 0.28, 0.56, 1.1, 2.8, 5.6, 17, and 33 mM.

## 4. Conclusions

By strategically designing the position of a lysine residue for PES modification by the structural comparison between GOx and FADGDHs, an engineered *An*GOx-I489K was constructed. *An*GOx-I489K was readily modified by arPES, which enables the enzyme to exhibit MTT system-dependent dye-mediated glucose dehydrogenase activity and an electrochemical response on the electrode upon the addition of glucose. These properties were distinct from those of PES-modified *An*GOx-WT. The enzyme electrode with PES-modified *An*GOx-I489K showed a response at a potential as low as 0 mV vs. Ag/AgCl, which enabled the elimination of an effect from electrochemically-active interference. Additionally, since a direct electron transfer-type sensor does not require a free electron mediator, we are able to simplify the architecture of a sensor.

In conclusion, the engineered GOx we show in this case has the advantage of quasi-DET-ability in addition to its original functionality (high specificity and activity). This achievement realized a revived version of the “gold standard” glucose sensing enzyme, GOx, which has been used in the first-generation and second-generation principles based on glucose sensors, which will be available for the “2.5th generation” based glucose sensors. The addition of quasi-DET ability to GOx make this enzyme the invaluable material to construct a mediator-free type glucose sensor strip, and an ingredient impact-free CGM sensor.

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
