# Peer review of "Engineered Glucose Oxidase Capable of Quasi-Direct Electron Transfer after a Quick-and-Easy Modification with a Mediator"

_ijms, 2020, doi:10.3390/ijms21031137_

Round 1

Reviewer 1 Report

Reviewer’s Reply

The author presented a modified glucose oxidase with a mediator to establish a sensor system for glucose detection. The background introduction, experiment design and data discussion is ample and sound. Some of the questions are listed below:

Line 52: “the signal derived” could be better explained by “the loss of the electrons from glucose oxidation…” Line 57: “DET-based measurements are achieved with lower potential than that of the first- or second-generation principles” Does the author have any reference for that? Line 66: the author mentioned “quasi-DET” in the manuscript. It could be improved by introduce the definition of “quasi-DET” if possible. Line 98: “However, when bulky MTT is used…this is the case for GOx” Does the author know the reason for that? Line127: “However, there is one notable residue that is located at what appears to be the … center” Would modification of those residues that close to the entrance of the pathway affect the binding of the enzymes with their substrates? Does the author have any other justification?

Author Response

Reviewer 1

R1-1 Line 52: “the signal derived” could be better explained by “the loss of the electrons from glucose oxidation…”

According to your advice, we have amended the sentence in line 54.

R1-2 Line 57: “DET-based measurements are achieved with lower potential than that of the first- or second-generation principles” Does the author have any reference for that?

Thank you for the important suggestion. We revised the sentence explaining the advantages of third-generation principle with lower potential in line 20, and line 54 to 61. According to your suggestion, we added references to show the redox potential of mediators used in second-generation principle of glucose sensor (ref.8; Chaubey et al.), and the applied potential of third-generation principle based glucose sensors (ref.9; Lee et al.).

R1-3 Line 66: the author mentioned “quasi-DET” in the manuscript. It could be improved by introduce the definition of “quasi-DET” if possible.

We appreciate for your kind suggestion. To define, we have added a sentence in line 81 to 85.

R1-4 Line 98: “However, when bulky MTT is used…this is the case for GOx” Does the author know the reason for that?

We previously reported that MTT alone did not function as the electron acceptor for variety of redox enzymes, however the presence of PMS facilitated the reduction of MTT by the oxidation substrate (Hatada et al.). Considering  that both PMS and MTT are negatively charged in their oxidative status, we assumed that the low reactivity of MTT as the primary electron acceptor might be attributed to its bulkiness compared with PMS. Therefore, we have used the availability of MTT as the sole electron acceptor in the solution as the indication that PES was successively modified on the surface of  redox enzymes, where internal electron transfer from deeply buried cofactor to PES occurred, consequently bulky MTT can be reduced. Actually, when we modified FADGDH with PES, the availability of MTT as the sole electron acceptor was significantly improved, and quasi-DET with electrode was also observed (Hatada et al.). We revised the description about the assumption of arPES modification of GOx and the evaluation method in line 105-116.

R1-5 Line127: “However, there is one notable residue that is located at what appears to be the … center” Would modification of those residues that close to the entrance of the pathway affect the binding of the enzymes with their substrates? Does the author have any other justification?

   According to your comments, we added the results of activity measurement for PES-modified GOx in the Table S1. Regarding dehydrogenase activity, the Km value of AnGOx-I489K before and after PES modification were 22 and 23 mM, respectively, and no significant change was observed. Therefore, it appears that the attachment of PES to the entrance did not altered the accessibility of glucose to the active center. Also, we added this discussion in line 165 to 168.

Reviewer 2 Report

This manuscript on "Engineered glucose oxidase capable of quasi-direct electron transfer after quick and easy ,modification with a mediator" is a novel and interesting piece of scientific information.The authors include renowned academics who have vast experience and knowledge of publishing in internationally peer reviewed journals and are also pioneers in this field.

I would like to raise the following concerns:

Kindly change the "Introduction" opening sentence as it is similar to that of the "Abstract". Please replace figure 1 with a figure that explains "Quasi-Direct Electron transfer"; this will benefit and interest readers with limited knowledge on the various generations of glucose biosensing. Kindly check for typos in the manuscript. Could figure 4 be presented as a dose response curve with Michealis -Menten fit? This will probably explain the enzyme kinetics involved.  In your conclusion, could you add some perspective as to how this technology will be finally adopted ( for example in the case of continuous glucose monitoring sensors).

Author Response

Reviewer 2

R2-1 Kindly change the "Introduction" opening sentence as it is similar to that of the "Abstract".

   We apricate for your kind suggestion. We have revised the abstract in line 16-17 and 19-20.

R2-2 Please replace figure 1 with a figure that explains "Quasi-Direct Electron transfer"; this will benefit and interest readers with limited knowledge on the various generations of glucose biosensing.

   Thank you very much for the valuable suggestion. Because the chemical reaction indicated in figure 1 is primary requirement by the editor. Therefore, keeping this figure as scheme 1, we added a new scheme, showing the principle of each generation of glucose sensor to explain Quasi-Direct Electron transfer as scheme 2.

R2-3 Kindly check for typos in the manuscript.

   We appreciate for the kind advice. We carefully checked again the entire manuscript.

R2-4 Could figure 4 be presented as a dose response curve with Michealis -Menten fit? This will probably explain the enzyme kinetics involved.

   We are afraid that lack in the explanation might resulted this comment. The results shown in Figure 5 in the former version was derived from the time course of the chronoamperometric measurement (former Figure 4). Therefore, the former Figure 5 corresponds to the Michealis -Menten plot of glucose response shown in former Figure 4. To avoid further confusion, we combined the former Figure 4 and 5 into one figure (Figure 3A and B in the revised version). Accordingly, we described the apparent Km value obtained from the electrode response in line 190.

R2-5 In your conclusion, could you add some perspective as to how this technology will be finally adopted (for example in the case of continuous glucose monitoring sensors).

   We appreciate for your suggestion. We added the perspective which could be achieved with the quasi-DET type GOx in line 225 to 229.

Reviewer 3 Report

The manuscript presents the rational modification of a GOx mutant with a phenazine ethosulfate mediator for providing a quasi-DET behavior at GOx modified electrodes. The presented results are interesting and could be potentially considered for publication after attending the following comments:

As summarized in Table 1, a considerable dehydrogenase activity has been observed for the GOx mutant including PES modification in the presence of MTT. However, it is surprising that the activity is only the same as for the wild type in the presence of PMS/MTT. This control should be repeated or a further detailed explanation provided.

What is the reason for a decrease in the reduction peak observed at around -0.4 V for all electrodes in the presence of glucose (Figure 3 and Figure S1, compare curves b, d with a, c)?

There seems to be a disagreement between the Km values reported in Table S1 and the dependence on glucose concentration for modified electrodes (Figures 4 and 5). Even when the values reported in Table S1 were estimated with enzyme in solution instead of immobilized on the electrode surface, the difference is about 10 times. Can the authors please comment on that?

Additional comments:

Figure 1. Since glucose is already included in the equation, the term “glucose” above the arrow is misleading and should be therefore removed.

Author Response

Reviewer 3

R3-1 As summarized in Table 1, a considerable dehydrogenase activity has been observed for the GOx mutant including PES modification in the presence of MTT. However, it is surprising that the activity is only the same as for the wild type in the presence of PMS/MTT. This control should be repeated or a further detailed explanation provided.

Thank you very much for your comment. In the MTT system using PES-modified GOx, the modified PES will simply replace the role of PMS in PMS/MTT system. Therefore, we expected the PES-modified GOx mutant would show similar MTT mediated  dehydrogenase activity as the WT GOx, and we do not have hypothesis that the PES modification shall increase the activity using MTT as the sole electron acceptor. We revised the description of our expectation and hypothesis in line 105-116.

R3-2 What is the reason for a decrease in the reduction peak observed at around -0.4 V for all electrodes in the presence of glucose (Figure 3 and Figure S1, compare curves b, d with a, c)?

Thank you for the valuable question. The reduction peak was attributed to the reduction of oxygen, and the decrease of the peak was caused by the consumption of oxygen by glucose oxidase on the electrode. We added this discussion in line 173 to 175.

R3-4 There seems to be a disagreement between the Km values reported in Table S1 and the dependence on glucose concentration for modified electrodes (Figures 4 and 5). Even when the values reported in Table S1 were estimated with enzyme in solution instead of immobilized on the electrode surface, the difference is about 10 times. Can the authors please comment on that?

Thank you for the valuable comment. The difference between Km values in Table S1 and the observed glucose concentration dependent current increase were derived from the difference of the rate limiting step. In the homogeneous solution, the reductive half reaction is set as the rate limiting step in the presence of enough concentration of electron acceptor in the solution to determine the Km value of the substrate (glucose). However, the results shown in Former Figure 4 and 5, now Figure 3A and B in the revised version, apparent Km value observed from the current increase was much lower. This result indicated that in the electrochemical measurement, the electron transfer between quasi-DET GOx and electrode was limited. Namely, the rate limiting step in the electrochemical reaction with PES modified GOx was assumed at the oxidative half reaction where reduced PES was oxidized by the electrode. Therefore, the apparent Km value observed on the electrode did not match the actual Km value determined with the redox dye.  We added the apparent Km value observed in the electrochemical assay  and added this discussion in line 190 to 198.

Additional comments:

R3-5 Figure 1. Since glucose is already included in the equation, the term “glucose” above the arrow is misleading and should be therefore removed.

Thank you for your kind suggestion. We have removed glucose above the arrow.

Round 2

Reviewer 3 Report

The manuscript has been considerably improved after the round of revisions. All my concerns and questions have been adequately answered. I recommend acceptance of the manuscript for publication.

Author Response

R1-1: (x) English language and style are fine/minor spell check required.

We have checked entire manuscript and amended.
